# Outcome Expectations for Exercise and Decisional Balance Questionnaires Predict Adherence and Efficacy of Exercise Programs in Dialysis Patients

**DOI:** 10.3390/ijerph17093175

**Published:** 2020-05-02

**Authors:** Špela Bogataj, Maja Pajek, Jadranka Buturović Ponikvar, Jernej Pajek

**Affiliations:** 1Department of Nephrology, University Medical Centre, 1000 Ljubljana, Slovenia; spela.bogataj@kclj.si (Š.B.); jadranka.buturovic@kclj.si (J.B.P.); 2Faculty of Sport, University of Ljubljana, 1000 Ljubljana, Slovenia; maja.pajek@fsp.uni-lj.si; 3Faculty of Medicine, University of Ljubljana, 1000 Ljubljana, Slovenia

**Keywords:** adherence, functional training, intradialytic cycling, outcomes expectations for exercise, decisional balance, exercise adoption, behavior change, questionnaire, hemodialysis

## Abstract

The purpose of this study was to define if Outcomes Expectations for Exercise (OEE) and Decisional Balance (DB) scales predict adherence to guided exercise programs and associate with the improvement in physical performance in the dialysis population. Participants (*n* = 40; age 63.6 ± 12.5 years) completed OEE and DB questionnaires before randomization to the experimental group (*n* = 20) and control group (*n* = 20) of a two-phased exercise program—the experimental group received eight weeks of supervised functional exercise and exercise counseling (1st phase) before commencing eight weeks of home-based exercise on non-dialysis days (2nd phase). Both groups performed intradialytic cycling on dialysis days during both study phases. Patients with above-median OEE and DB scores (>3.15 and >1.3, respectively) expressed significantly better adherence to intradialytic cycling (89% vs. 76%, 89% vs. 77%, respectively, *p* < 0.05). Experimental group patients with an above-median OEE (but not DB) score had significantly better adherence to supervised and home-based functional exercise (93% vs. 81% and 85% vs. 60%, respectively, *p* < 0.05). Baseline DB score predicted the final result in the hand-grip test and 6-min walk test. Low OEE and, to a lesser degree, low DB questionnaire scores associate with inferior adherence to dialysis bundled and home-based exercise programs and may help define patient subsets in need of intensified motivational input by exercise caregivers.

## 1. Introduction

Chronic kidney disease (CKD) has become a public health problem that affects 8%–16% of the population worldwide [1]. CKD patients typically lead very sedentary lifestyles [2], which contributes to increased risk of cardiovascular disease [3], decreased physical function [4,5,6], muscle wasting [7], and overall reduced quality of life [8]. Hemodialysis (HD) is one of the main methods of renal replacement therapies that are employed to treat advanced kidney disease [9]. Studies have identified several barriers in HD patients that significantly affect exercise adherence [2,10]. Although, generally, little is known about the psychological factors that limit exercise adoption in HD patients, lack of motivation and interest are the two recognized factors that limit patient participation in physical activity [11].

To increase the success of physical exercise programs in the dialysis population, we need tools to predict adherence to these programs and establish patients’ attitudes towards organized physical activity. According to Pinto et al. [12], decisional balance (DB) is a psychological construct that has undergone much empirical research. The DB, as defined by Plotnikoff [13], involves the perceived advantages (pros) and disadvantages (cons) of continuing a current behavior or adopting a new behavior. However, besides DB, some studies [14,15] reported that the Outcome Expectations for Exercise (OEE) scale is considered as a significant predictor of exercise behavior. Outcome expectations represent the belief that carrying out a specific behavior will lead to the desired outcome [15]. DB and OEE questionnaire tools could measure and represent a critical aspect of an individual’s decision to commit to exercise. 

It is well documented that behavior change interventions are drawn from psychological models of human behavior [16]. Samdal et al. [17] support the use of goal setting and self-monitoring of behavior when promoting change in physical activity. Furthermore, Michie et al. [18] have provided a significant basis and evidence for physical activity interventions showing the specific links between behavioral change techniques and theoretical constructs. The effectiveness of social cognitive theory in exercise research was examined in a variety of patient populations [19]. There is a lack of studies examining the changes of behavior associated with exercise adoption in hemodialysis patients, and relatively little empirical research has been conducted on the relations of OEE and DB to exercise adherence. According to Anding et al. [20], the patient’s adherence is crucial for any intervention to be effective. Considerable evidence demonstrates the attrition level for exercise interventions in CKD to be around 30%−40% [10,21,22,23,24,25,26,27,28]. Therefore, we aimed to define if a baseline score on the OEE and DB scale can predict the adherence and success of dialysis patients in an organized exercise program, which of the two instruments performs better, and how do they perform in monitoring the change of attitudes toward exercise. The baseline and subsequent change in patients’ attitudes toward exercise using OEE and DB questionnaires was measured after 8 and 16 weeks of participation in a randomized controlled trial of two dialysis-bundled exercise programs. It was hypothesized that higher results for OEE and DB questionnaires associate with higher adherence with and larger improvement of physical exercise intervention.

## 2. Materials and Methods

Prevalent HD patients were recruited at hemodialysis units of the University Medical Centre in Ljubljana, Slovenia, to participate in a trial comparing the effects of two exercise programs on their physical performance. They were randomized into the functional exercise group (experimental group; *n* = 20), and the intradialytic cycling group (control group; *n* = 20). Physical performance (regarded here as a multidimensional concept including lower-extremity function (mobility), upper-extremity function (dexterity), and capability to carry out activities of daily living) was measured objectively with functional physical performance tests [29,30,31]. We measured lower limb strength with 10 repetitions of the sit-to-stand test [32], the aerobic capacity by the six-minute walk test [33], and hand-grip strength [34] assessed with a calibrated hydraulic hand dynamometer (JAMAR 5030 J1, Patterson Medical, Warrenville, Ilinois) at baseline, after eight weeks, and after 16 weeks of the intervention. The data hereby presented were gathered in a randomized controlled trial comparing functional exercise combined with intradialytic cycling vs. intradialytic cycling only in dialysis patients with the exact patient selection process, exercise interventions, protocol, and patient flow described in detail previously [31]. The experimental group engaged in guided functional exercise before each HD session (predialysis functional exercise) and intradialytic cycling exercise (during dialysis). The functional exercise simulates activities of daily living [35] and targets the neuromuscular system to train movements that activate the nervous system and the muscle groups [36]. Functional exercise is performed as a combination of lower and upper body movements, along with various multi-joint activities [37]. The functional exercise in our study lasted for up to 30 min before each HD session and was guided by the kinesiologist. The number of repetitions, sets, and the load was individualized to achieve the desired intensity of the rate of preserved exertion (RPE) of 7th to 8th grade on a modified Borg scale (range 0 to 10). We started with about five different exercises with ten repetitions in two sets without the extra weight. Then, we gradually increased the number of repetitions, or we added weight. We aimed to achieve the completion of three sets of each exercise with 10–15 repetitions. The warm-up consisted of light cardiovascular exercises and coordination and balance exercises. The main part of the functional exercise consisted of varieties of squats, lunges, push-ups, pulls, pushes, and lifts tailored to each individual’s abilities. The cool-down included light cardiovascular exercises alongside stretching. The exact exercise content is additionally described in ref. [31]. ntradialytic cycling was performed in the first two hours of HD procedure. It was supervised by the same kinesiologist who aimed to continuously progress the cycling resistance or time to maintain the RPE of 4th to 5 th grade on a 10-grade Borg scale. Initial intradialytic cycling duration was set to 15 min with a gradual increase to reach the duration of up to 60 min. An increase in resistance and duration was individualized according to each patient’s RPE response and their motivation. This exercise prescription strategy was the same for the experimental and control group.

After eight weeks, a predialysis functional exercise was terminated, and patients were instructed and motivated to perform functional exercise routines at home on non-dialysis days for an additional eight weeks (home-based functional exercise). On dialysis days, intradialytic cycling was continued. The control group performed intradialytic cycling only in both study periods (16 weeks). 

### 2.1. Study Criteria And Ethical Considerations

Inclusion, exclusion, and withdrawal criteria are presented in Table 1. National Medical Ethics Committee approval was attained from the Ministry of Health, Republic of Slovenia (approval document number 0120–97/2017–3 KME 68/03/17). Written informed consent was signed from all subjects, and information about the aim, confidentiality, and procedures of the study was explained. The study complies with the Declaration of Helsinki (2013). The study was registered at ClinicalTrials.Gov on the 7th of November 2017 (Clinicaltrials.gov identifier: NCT03334123).

### 2.2. Study Design 

The study timeline is presented in Figure 1. We compared two different exercise approaches, (1) functional exercise with exercise counseling in addition to intradialytic cycling and (2) solely intradialytic cycling. Additionally, experimental group patients were transferred to an unsupervised, home-based functional exercise in the second phase of the study. They were advised, monitored, and motivated to perform the functional exercise at home, three times a week. On dialysis days, we assessed adherence and discussed the issues of home functional exercise. We gave them a written individualized exercise program with exercise descriptions and illustrations and discussed how to perform them in their home environment. At each dialysis session of study phase 2, they reported the details of the exercise performed. The exercise was counted as completed if they maintained the same RPE level and implementation of suggested exercises as in the first guided phase of the study. 

Psychological questionnaires, body composition, and physical performance tests were assessed at baseline, after eight weeks and after 16 weeks. Decision Balance scale (DB), Outcome Expectations for Exercise scale (OEE), body mass index (BMI), bioimpedance derived lean tissue index (LTI), and fat tissue index (FTI, Body Composition Monitor, Fresenius AG, Bad Homburg, Germany), 6-min walk test (6MWT), 10 repetitions sit-to-stand test (STS10), and hand-grip strength test (HG) were the assessed outcomes. The same assessors were assigned to each independent endpoint at all measurements and were blinded to treatment allocation. Questionnaires were read out loud to participants to make sure that they understood the question/statement.

### 2.3. The Main Study Outcomes

#### 2.3.1. Decisional Balance Scale

The participants’ perceptions about the pros and cons of physical exercise were measured with the DB scale [38]. The questionnaire consists of 16 items. Advantages (pros) are defined with ten items, whereas six items reflect the disadvantages (cons) of participating in physical exercise. The pros subscale includes the following items: “I would have more energy for my family and friends if I exercised regularly”; “Regular exercise would help me relieve tension”; “I would feel more confident if I exercised regularly”; “I would sleep more soundly if I exercised regularly”; “I would feel good about myself if I kept my commitment to exercise regularly”; “I would like my body better if I exercised regularly”; “It would be easier for me to perform routine physical tasks if I exercised regularly”; “I would feel less stressed if I exercised regularly”; “I would feel more comfortable with my body if I exercised regularly”; “Regular exercise would help me have a more positive outlook on life”. The items in the cons subscale are: “I think I would be too tired to do my daily work after exercising”; “I would find it difficult to find an exercise activity that I enjoy that is not affected by bad weather”; “I feel uncomfortable when I exercise because I get out of breath and my heart beats very fast”; “Regular exercise would take too much of my time”; “I would have less time for my family and friends if I exercised regularly”; “At the end of the day, I am too exhausted to exercise”. Answers are given on a Likert-type scale from 1 (not important) to 5 (very important). The decisional balance scale score was calculated by subtracting the cons factor mean from the pros factor mean. This way, the higher DB score reflects a stronger belief in the positive benefits of an exercise. The reliability of the DB was calculated to have a Cronbach α of 0.85 in the present study.

#### 2.3.2. Outcome Expectations for Exercise Scale

Physical and mental health perceptions about exercise were measured using the OEE scale [15], which focuses on the benefits and outcomes expectations associated with exercise. The OEE scale provides nine statements about exercise for evaluation on a scale from 1 (strongly disagree) to 5 (strongly agree). The following nine statements were included: “Makes me feel better physically”; “Makes my mood better in general”; “Helps me feel less tired”; “Makes my muscles stronger”; “Is an activity I enjoy doing”; “Gives me a sense of personal accomplishment”; “Makes me more alert mentally”; “Improves my endurance in performing my daily activities”; and “Helps to strengthen my bones”. The final score was calculated by summing the ratings of each response and dividing this number with the number of all responses (nine). The reliability was confirmed in our study with a Cronbach α of 0.92. 

#### 2.3.3. Patient Adherence

Adherence to exercise programs were defined as the total number of completed exercise sessions in contrast to the total number of sessions offered. The formula was:Patient’s adherence (%) = number of completed sessions/number of offered sessions ∗ 100


### 2.4. Statistical Analysis

Descriptive statistics (mean ± SD) were calculated for participants’ demographic and clinical characteristics. The reliability for OEE and DB was determined using Cronbach α with the following formula:
(1)$$\alpha =\frac{N*\bar{c}}{\bar{v}+(N-1)*\bar{c}}$$
N = the number of subjects; $\bar{c}$ = average covariance between pairs; $\bar{v}$ = average variance.

An independent-samples t-test was used to compare the group’s baseline characteristics. A paired t-test was used to analyze within-groups changes over time in questionnaire scores. Between-group differences were tested with analysis of covariance. Cohen *d* effect sizes (ES) were calculated to determine the magnitude of the group differences in OEE and DB scores. ES was classified as: <0.2 was defined as trivial, 0.2 − 0.6 as small, 0.6 − 1.2 as moderate, 1.2 − 2.0 as large, and >2.0 as very large [39]. An independent samples t-test was used to check for the differences in exercise adherence, where subjects were divided into two subgroups according to their score on the questionnaire: a low score subgroup (OEE or DB score ≤ median) and a high score subgroup (OEE or DB score > median). Division by the median was chosen to avoid any investigator-imposed bias in the choice of the division threshold. General linear models were used to determine associations between the OEE and DB baseline scores, baseline physical performance tests results (6MWT, STS10, and HG) and body composition indices (BMI, LTI, and FTI). We examined associations between the OEE and DB baseline scores and improvement after 16 weeks in physical performance tests by using analysis of covariance, where we set the final physical performance test result as a dependent variable, and the baseline test result and questionnaire baseline score as covariates. All tests were two-sided, carried out using SPSS, version 22 (SPSS Inc., Chicago, IL, USA), and assessed at the *p* < 0.05 level of significance. Since this was a first exploratory analysis of OEE and DB questionnaire performance as exercise adherence predictors in a limited sample, no *p*-value adjustments for multiple comparisons were made. 

## 3. Results

We were able to include 40 prevalent HD patients in this trial; subsequently, 16 patients in experimental and 18 patients in the control group successfully finished both study phases. Patients’ baseline demographic and clinical characteristics are presented in Table 2. 

Baseline scores for both questionnaires are shown in Figure 2. The median for the OEE scale was 3.15 points (interquartile range = 0.9) and for DB 1.3 points (interquartile range = 1.6). No significant differences in age, sex distribution or dialysis vintage between high and low OEE and DB questionnaire groups could be found. 

We analyzed associations between possible predictive parameters for the baseline result in OEE and DB scale by utilizing the univariate general linear model (Table 3). OEE and DB in this model are dependent variables, whereas 6MWT, STS10, HG, BMI, LTI, and FTI represent independent predictor variables. The model results showed that the OEE baseline score is significantly positively associated with a baseline result on a 6MWT (*p* = 0.001), in HG (*p* = 0.024) and with the baseline LTI (*p* = 0.033). This means that better functioning at baseline and higher LTI were associated with a higher OEE score. Baseline DB performance was not associated with any physical performance or body composition indices. 

The progression of changes of attitudes to exercise as captured by OEE and DB questionnaires is presented in Figure 3 by showing baseline, end of phase 1 (after eight weeks), and end of phase 2 (after 16 weeks) results. Mean ± SD values and effect sizes are presented in Table 4. 

There were no significant differences between the groups at baseline. There was a significant improvement in OEE and DB scores after 16 weeks in the experimental group (OEE: 1.21 ± 0.71 (CI 95% 0.83 to 1.59; *p* < 0.001); DB: 0.9 ± 0.93 (CI 95% 0.41 to 1.39; *p* = 0.001)) and significant improvement in the OEE score of 0.72 ± 0.64 (CI 95% 0.4 to 1.04) in the control group. There was a significant between-group difference at the end of the first (*p* = 0.022) and second (*p* = 0.022) study phase in the OEE score. In the DB scale, there were no significant between-group differences. 

For the main study endpoint—adherence analysis—the subjects were divided into a high and a low questionnaire score subgroups divided by the median central tendency (see Figure 2). Table 5 presents differences in adherence to exercise programs in the subgroup with a high baseline score (scores above median) and in the subgroup with a low baseline score (scores below median) in both questionnaires. Patients were divided twice, according to baseline OEE, and to DB median score (Figure 2). In the OEE high group, there were nine patients from the EXP group and 11 patients from the CON group. The OEE low group consisted of 11 EXP and 9 CON patients. In the DB high group, there were 12 patients from the EXP group and ten patients from the CON group. In the DB low group, there were 8 EXP patients and 10 CON patients. There was a significant difference between the high OEE score subgroups and low OEE score subgroups in adherence to cycling exercise, predialysis functional exercise, and home-based functional exercise. The difference between high DB score and low DB score subgroups was significant for the adherence to cycling exercise only.

A possible predictive value of baseline OEE and DB scores for the final achieved physical performance result was tested by analyzing the association of baseline OEE and DB scores with the final physical performance parameters adjusted for the baseline physical performance results (Table 6). 

The final physical test result was set as a dependent variable adjusted for its baseline value with OEE and DB baseline results set as an independent predictor variable. 

The analysis showed that the baseline DB score positively predicted the achieved performance after 16 weeks in the HG test (*p* = 0.013), 6MWT (*p* = 0.009), and a trend to significant association with improvement in the STS10 result (*p* = 0.062). No significant associations with the final achieved physical performance parameters were found for baseline OEE questionnaire score.

## 4. Discussion

Successful implementation of an exercise program for HD patients is largely dependent upon the factors influencing exercise participation [40]. To date, only isolated studies can be found in dialysis patients that support the use of social cognitive theory in understanding physical activity and exercise outcomes [41]. The present study aimed to address the gap in the empirical literature concerning the prediction of exercise adherence using specific questionnaires in the dialysis patient population. By examining the association of Outcome Expectations for Exercise and Decisional Balance scales with adherence to exercise programs in HD patients, the main findings of this study are that the score on the OEE scale predicted exercise adherence and did this comparatively better than the DB scale. Higher baseline levels of the OEE scale were associated with higher exercise adherence 8 and 16 weeks after the implementation of both (higher and lower volume) exercise modalities. Therefore, we can expect that exercise adherence could be improved if potential expectations for exercise were determined and patients with poor results (e.g., below 3.15 as in our study) given additional motivational attention by caregivers and trainers. This way, OEE results can give us an insight into how willing the dialysis patient is to exercise and how much effort and motivational techniques we need to put into each individual. When examining the relations between baseline questionnaire scores, body composition indices and physical performance, we found significant univariate associations between LTI, HG, 6MWT, and OEE scores, suggesting that relatively healthier and physically stronger patients exert better OEE results. In part, this may be one of the reasons for better prediction of exercise adherence by higher OEE scores. 

The evolution of patients’ attitudes during participation in exercise programs showed positive changes with both scales; however, the OEE score improved in experimental and control groups, while DB score improved in the experimental group (with a larger volume of exercise and larger counseling input) only. An initial improvement in OEE and DB scores after the first study phase is not surprising given that individuals without experience (as was the case with our participants prior to entering this study) are unable to accurately rate their confidence for a task as they often overestimate or underestimate their capabilities [42]. However, the improvement in OEE scores between and within groups remained significant during the transition from a supervised to a self-managed, home-based exercise in the second phase and we speculate that better results for the experimental group during the first phase of the intervention contributed to the development of a stronger sense of mastery through the practice of home-based functional exercise. Concerning the DB scale, patients in the experimental group have increased their pros and decreased the cons after exercise participation. However, there were no improvements for the control group in DB scale results at 8 or 16 weeks, suggesting that the control intervention was not effective in changing the pros and cons of exercise. It can be speculated that the differences in DB scale results between groups did not reach statistical significance partly due to the relatively large variability in scores and relatively low patient numbers. 

It is well documented that the perceived exercise benefits and barriers to exercise are related to the exercise behavior of individuals [43]. Accordingly, Kao et al. [44]. suggested that behavior modification could increase levels of exercise in CKD patients. Michie et al. [18] stated that identifying barriers and finding ways to overcome them is a popular technique used in behavior change interventions. Additionally, interventions prompting participants to self-monitor their behavior were more effective in achieving behavior change [18]. Pinto et al. [12] stated that individuals who are in more advanced stages of motivational readiness for exercise report more positive DB adoption. However, it is not fully understood whether DB and OEE are differentially associated with physical activity under different conditions (e.g., engaging in exercise for rehabilitation vs. for general health) and among different populations (e.g., diseased vs. healthy). For instance, in rehabilitation settings in which physical activity plays an important role, physical outcome expectations may be more salient in the initial stages of the rehabilitative process.

Although baseline DB score proved inferior to OEE in predicting adherence, it was significantly associated with the improvement after 16 weeks in the HG test and 6MWT. Although there is clear evidence that exercise interventions can increase exercise behavior and improve clinical outcomes among patients with chronic diseases [10,44,45,46], to date, there has been little clinical interest in elucidating the role of OEE and DB in exercise programs effectiveness, especially in HD patients. One study that used OEE and DB conducted on depressive diabetic patients showed that based on their results, they might require additional support for exercise initiation and maintenance [47]. Since the physical performance of dialysis patients may be relatively less dependant on uremic toxicity [48], a greater input into the study of social and psychological factors associated with exercise adoption is needed in this patient population. We believe that constant personal contact of the kinesiologist and nurses combined with strong support from physicians was the key to high exercise adherence in our study. Additionally, close family members were informed about the study and welcomed to support the patient.

Although the findings of the present study are encouraging, there are several limitations. One limitation which is common in psychological research is that the scores representing behavioral changes were based on self-report. Moreover, the small sample size could be considered as a limitation, although we did find some clearly significant results. Specifically, the thresholds for division into high and low OEE and DB questionnaire scores may not be generalizable to other, especially non-dialysis populations. As this was a first exploratory analysis of these questionnaires for adherence prediction, usual precautions about external generalization of these results must be considered regarding not only the sample size but also regarding the nature of the sample, sex, age variability, and other sociodemographic variables that may be considered in future studies.

Therefore, it would be beneficial to investigate further the value of these questionnaires by increasing the sample size and conducting stratified sampling of patients from different regions and cultural backgrounds.

## 5. Conclusions

The present study is the first study to examine the relation of DB and OEE scales to adherence and performance of HD patients in structured exercise programs. We found that low OEE and, to a lesser degree, low DB questionnaire scores associate with inferior adherence to dialysis-bundled and home-based exercise programs. Additionally, OEE scores improved significantly during participation in high (experimental) and lower exercise volume (control) groups, while DB score improved in the experimental group with a much larger exercise and counseling input only. These findings on the utility of OEE and DB questionnaires to predict adherence and monitor exercise-related patient attitudes may help define patient subsets in need of intensified motivational and supervision input by exercise caregivers.

## Figures and Tables

**Figure 1 ijerph-17-03175-f001:**
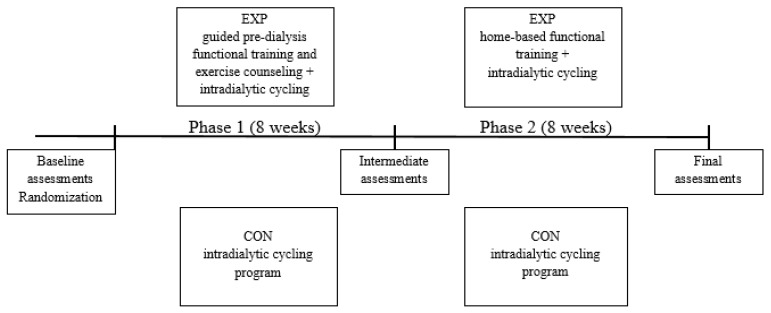
Study overview.Abbreviations: EXP, experimental group; CON, control group.

**Figure 2 ijerph-17-03175-f002:**
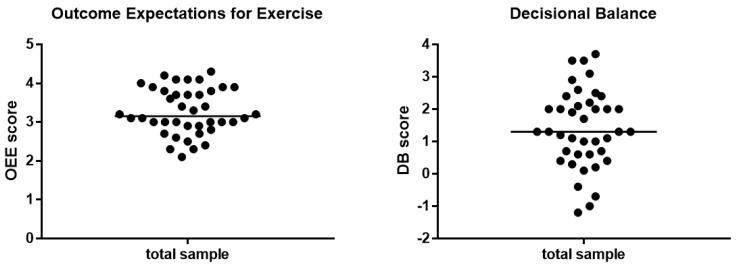
Baseline scores for OEE and DB scale divided by median. OEE: Interquartile range = 0.9; min-max = 2 − 4.3. DB: Interquartile range = 1.6; min-max = −1.2 − 3.5. Abbreviations: OEE, outcome expectations for exercise; DB, decisional balance.

**Figure 3 ijerph-17-03175-f003:**
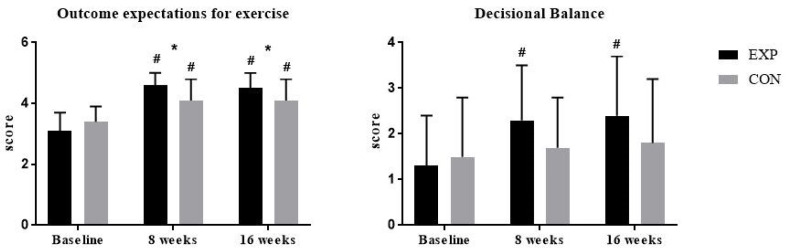
Within-group and between-group changes in questionaire scores. Abbreviations: EXP, experimental group; CON, control group. # *p* < 0.05 within-group change compared to baseline; * *p* < 0.05 for between-group difference at specific time points.

**Table 1 ijerph-17-03175-t001:** Study criteria.

Inclusion Criteria	Exclusion Criteria	Withdrawal Criteria
patients on HD renal replacement therapy > 3 months	chronic malignant or infectious disease	any intercurrent illness or trauma
18−90 years old	uncontrolled arterial hypertension	the occurrence of an acute illness
stable medical condition	angina pectoris of Canadian Cardiovascular Society grade 2−4	diagnosis of malignant disease during the research
capable of independent walking and feeding	New York Heart Association heart failure grade 3 or 4	withdrawal of the consent to participate in the research
	the presence of psychotic illness or a mental disability	
	a history of limb amputation (more than two fingers on the lower limb and/or upper limb)	

Abbreviations: HD, hemodialysis.

**Table 2 ijerph-17-03175-t002:** Demographic and clinical characteristics.

Variable	All Participants (*n* = 40)	Experimental Group (*n* = 20)	Control Group (*n* = 20)
Age (years)	63.6 ± 12.5	65.2 ± 12.1	61.9 ± 13.0
Male sex (%)	55%	60%	50%
Height (cm)	167.9 ± 9.8	168.4 ± 9.6	167.5 ± 10.2
Weight (kg)	72.1 ± 15.8	72.6 ± 16.1	71.7 ± 15.9
Dialysis vintage (years)	7.4 ± 7.7	7.4 ± 8.1	7.5 ± 7.3
Weekly dialysis duration (h)	12.9 ± 2.3	12.5 ± 2.7	13.3 ± 1.9
Lean tissue index (kg/m^2^)	13.3 ± 2.6	13.6 ± 3.2	12.9 ± 2.0
Fat tissue index (kg/m^2^)	11.5 ± 5.4	11.4 ± 4.8	11.6 ± 6.1
Phase angle (°)	5.0 ± 0.9	5.2 ± 0.9	4.7 ± 0.9
BIA assessed overhydration (L)	1.4 ± 1.9	0.9 ± 1.1	1.9 ± 2.4

Note: Values are expressed as mean ± SD, number of subjects (percent). There were no statistically significant differences between the groups. Blood pressure was defined as the mean of the last three predialysis blood pressure values. Phase angle measurements were performed with an 800 μA current at a frequency of 50 kHz. Abbreviations: *n*, number of subjects; BIA, bioimpedance performed using Body Composition Monitor, Fresenius AG, Bad Homburg, Germany.

**Table 3 ijerph-17-03175-t003:** Associations of baseline physical performance and body composition indices with baseline questionnaire scores.

Variable	OEE				DB			
	B	95% CI	*p*	Partial Eta Squared	B	95% CI	*p*	Partial Eta Squared
**6MWT**	0.006	2.97 ± 0.16 (2.64 to 3.29)	0.001	0.26	0.002	1.39 ± 0.17 (1.04 to 1.73)	0.257	0.03
**STS10**	−0.043	2.97 ± 0.18 (2.6 to 3.33)	0.079	0.08	−0.028	1.39 ± 0.17 (1.04 to 1.73)	0.224	0.04
**HG**	0.059	2.97 ± 0.18 (2.61 to 3.32)	0.024	0.13	−0.006	1.39 ± 0.18 (1.03 to 1.74)	0.799	0.002
**BMI**	0.034	2.97 ± 0.19 (2.59 to 3.34)	0.376	0.02	0.024	1.39 ± 0.18 (1.03 to 1.74)	0.517	0.01
**LTI**	0.150	2.97 ± 0.18 (2.61 to 3.32)	0.033	0.11	0.012	1.39 ± 0.18 (1.03 to 1.74)	0.864	0.001
**FTI**	−0.005	2.97 ± 0.19 (2.59 to 3.35)	0.891	0.001	0.014	1.39 ± 0.18 (1.03 to 1.74)	0.678	0.01

Abbreviations: OEE, outcome expectations for exercise; DB, decisional balance; 6MWT, 6-min walk test; STS10, 10-repetitions sit-to-stand test; HG, handgrip strength; BMI, body mass index; LTI, lean tissue index; FTI, fat tissue index; B, beta (unstandardized coefficient); CI, confidence interval. Associations were tested by the univariate general linear model whereby Y (dependant variable) = B (coefficient) × X (independent variable) + U (error); B, independent variable model coefficient; 95% CI, 95% confidence interval for B; *p*, statistical probability; partial eta squared, measure of effect size representing the proportion of total variance of dependant variable associated with the independent variable. These analyses were done in the total sample *n* = 40.

**Table 4 ijerph-17-03175-t004:** Results in outcome expectations for exercise and decisional balance with effect sizes.

Group	Outcome Expectations for Exercise					Decisional Balance				
	Baseline mean ± SD	8 weeks mean ± SD	ES	16 weeks mean ± SD	ES	Baseline mean ± SD	8 weeks mean ± SD	ES	16 weeks mean ± SD	ES
**EXP**	3.1 ± 0.6	4.6 ± 0.4	2.9	4.5 ± 0.5	2.5	1.3 ± 1.1	2.3 ± 1.2	0.9	2.4 ± 1.3	0.9
**CON**	3.4 ± 0.5	4.1 ± 0.7	1.2	4.1 ± 0.7	1.2	1.5 ± 1.3	1.7 ± 1.1	0.2	1.8 ± 1.4	0.2

Abbreviations: EXP, experimental group; CON, control group; SD, standard deviation; ES, Cohen’s deffect size.

**Table 5 ijerph-17-03175-t005:** Differences between adherence to exercise programs in the low and high group based on the baseline score in questionnaires.

**Adherence**	**OEE High Group**	**OEE Low Group**	***p***	**Mean Difference (95% CI)**
average adherence at cycling exercie (both study phases, *n* = 34)	89% ± 11%	76% ± 18%	0.012	13.0 ± 4.9 (3.1 to 22.9)
average adherence to supervised predialysis functional exercise (phase 1, *n* = 17) *	93% ± 7%	81% ± 13%	0.001	12.4 ± 3.4 (5.5 to 19.3)
average adherence to home-based functional exercise (phase 2, *n* = 16) *	85% ± 8%	60% ± 22%	<0.001	24.2 ± 5.3 (13.4 to 34.9)
	**DB High Group**	**DB Low Group**	***p***	**Mean Difference (95% CI)**
average adherence at cycling exercise (both study phases, *n* = 34)	89% ± 11%	77% ± 17%	0.018	11.6 ± 4.7 (2.1 to 21.1)
average adherence to supervised predialysis functional exercise (phase 1, *n* = 17) *	86% ± 15%	87% ± 9%	0.731	1.4 ± 4.0 (−6.8 to 9.6)
average adherence to home-based functional exercise (phase 2, *n* = 16) *	75% ± 20%	71% ± 22%	0.631	3.4 ± 7.0 (−10.9 to 17.7)

Note: OEE high/low group defined with OEE questionnaire scores above/below 3.15; DB high/low group defined with a DB questionnaire score above/below 1.3. Score values are presented as mean ± standard deviation; adherence to functional exercise is calculated for the experimental group only. Adherence to cycling exercise is calculated for all participants as they all participated in intradialytic cycling exercise. Abbreviations: OEE, outcomes expectations for exercise scale; DB, decisional balance scale; CI, confidence interval. * Calculated for the experimental group only.

**Table 6 ijerph-17-03175-t006:** Association between baseline OEE/DB score and physical improvement after 16 weeks.

Change after 16 weeks	OEE				DB			
	B	95% CI	*P*	Partial Eta Squared	B	95% CI	*p*	Partial Eta Squared
**6MWT**	17.71	−18.7 to 54.1	0.329	0.03	22.864	6.1 to 39.6	0.009	0.21
**STS10**	−0.53	−4.6 to 3.5	0.790	0.002	−1.84	−3.8 to 0.1	0.062	0.11
**HG**	0.122	−1.9 to 2.1	0.900	0.001	1.22	0.3 to 2.2	0.013	0.18

Note: Abbreviations: OEE, outcomes expectations for exercise scale; DB, decisional balance scale; 6MWT, 6-min walk test; STS10, 10 repetitions sit-to-stand test; HG, hand-grip strength test; B, beta (unstandardized coefficient); CI, confidence interval. N = 34 for all analyses (all patients successfully finishing both study phases included).

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
