# Peer review of "Outcome Expectations for Exercise and Decisional Balance Questionnaires Predict Adherence and Efficacy of Exercise Programs in Dialysis Patients"

_ijerph, 2020, doi:10.3390/ijerph17093175_

Round 1

Reviewer 1 Report

The manuscript titled “Outcome Expectations for Exercise and Decisional Balance questionnaires help predict adherence and efficacy of exercise programs in dialysis patients” investigated that whether adherence and efficacy of exercise programs in dialysis patients can be predicted based on OEE and DB. The results indicate that the relationships between OEE/DB and adherence and physical improvement.

The importance of investigating the relationships between psychological state and executions of exercise program. Experimental information such as the obtained data, properties of subjects and so on are also well described. However, there are unclear points in the paper and additional experiments are needed. Therefore, it is not acceptable for publishing at the current paper.

Comments are the following.

- It is considered that several different words are used for describing one thing. Please unify the words.

- Title: Please make it clear whether OEE and DB “help predict” or “predict”. The authors described that “…OEE scale predicted exercise adherence …” What is the difference between  “help predict” and “predict”?

- Page 2, line 66: Why did you define eight weeks as the span of phase? Please describe the reasonable explanation.

- Page 2, line 73: Please describe the explanation of “physical performance”.

- Page 2, line 74: Please describe the explanation of “functional exercise training”. Is it the same training against each subject? What is the condition of training completion?

- Page 2, line 75: Please describe the explanation of “additional intradialytic cycling exercise”. Is it the same training against each subject? What is the condition of training completion?

- Page 2, line 75: What is the “pre-dialysis functional training”?

- Page 2, line 76, Please describe the explanation of “functional exercise”.  Is it the same training against each subject?

- Table 2: Please describe the reason why exclusion criteria were excluded from the experiment.

- Page 4, line 75: Please describe the explanation of “home-based functional exercise”. Is it the same training against each subject? What is the condition of training completion?

- Page 5, 2.3: Please describe all questionnaires of DB and OEE to understand the experiment.

- Page 5, line 127: Please describe the formula of Cronbach alpha.

- Page5, line 138: The adherence was not the ratio but the total number? Formula helps readers understand.

- Page 5, line 158: Please describe the definition of “Absolute baseline scores”.

- Page 6, lines 167-169: Please describe the formula of the univariate general linear model to clarify the explanatory and objective variables.

- Table 3: Please explain B, 95% CI, p and Partial Eta Squared.

- Page 8, line 199: Does “cycling sessions” mean “additional intradialytic cycling exercise” in line 75? If so, please unify it.

- Page 8, line 199: Does “in-center functional training” mean “functional exercise training” in line 75? If so, please unify it.

- Page 8, lines 194-195: In the higher and lower group of OEE and DB, how was the composition ratios of the experimental and control group? Please describe it.

- Page 10, line 232: Please describe how the authors concluded that OEE scale can be used for predicting the exercise adherence. Does predicting the exercise adherence mean up/down of exercise adherence or the total number (the definition of adherence described in Page 5) ? How was the error of prediction?

- As described in lines 252-260, the intervention might contribute to the development of a positive psychological state such as a stronger sense of mastery. The positive psychological state might be correlated to OEE/DB and adherence. But the intervention might also contribute to the improvements in physical performance. Even if there were  correlations between OEE/DB, adherence and physical performance, these correlations might be caused by the intervention. It is difficult to judge OEE/DB could be used for predicting physical performance because the effect of the intervention can not be ignored. The further experiment is needed.

Author Response

Reviewer 1

The manuscript titled "Outcome Expectations for Exercise and Decisional Balance questionnaires help predict adherence and efficacy of exercise programs in dialysis patients" investigated that whether adherence and efficacy of exercise programs in dialysis patients can be predicted based on OEE and DB. The results indicate that the relationships between OEE/DB and adherence and physical improvement.

The importance of investigating the relationships between psychological state and executions of exercise program. Experimental information such as the obtained data, properties of subjects and so on are also well described. However, there are unclear points in the paper and additional experiments are needed. Therefore, it is not acceptable for publishing at the current paper.

Our response: Thank you for your detailed review of our manuscript and for providing some insightful and thought-provoking suggestions to strengthen our manuscript. We feel we have sufficient responses to each of your major concerns listed above, which are further detailed below, and hope that they alleviate the concerns you have regarding the approaches adopted in our manuscript.

  1. REMARK- It is considered that several different words are used for describing one thing. Please unify the words.

Our response: We unified the words: functional exercise/training into only functional exercise; pre-dialysis/predialysis to only predialysis; cycling sessions/exercise to only cycling exercise: in-center functional training/functional training exercise to only functional exercise.

  1. REMARK- Title: Please make it clear whether OEE and DB "help predict" or "predict". The authors described that "…OEE scale predicted exercise adherence …" What is the difference between "help predict" and "predict"?

Our response: In our study, the questionnaires predicted the adherence to implicated exercise programs, so we changed the title accordingly. Thank you for noticig that. It is now changed to: »Outcome Expectations for Exercise and Decisional Balance questionnaires predict adherence and efficacy of exercise programs in dialysis patients«

  1. REMARK - Page 2, line 66: Why did you define eight weeks as the span of phase? Please describe the reasonable explanation.

Our response: Eight weeks was recognized as the minimum amount of time to expect exercise benefits, so that is why we choose an intervention length of eight weeks for the first phase and eight weeks for the second phase. A recent meta-analysis by Huang et al.[1] revealed that aerobic exercise or combined exercise at least for eight weeks to 12 months, three times weekly, was beneficial to the physical conditions of hemodialysis patients. So the eight week period was chosen to provide the minimal time which is evidence based to secure the possible measurable impact of exercise intervention.

  1. REMARK - Page 2, line 73: Please describe the explanation of "physical performance".

Our response: Physical performance is a multidimensional concept, which includes lower extremity function (mobility), upper extremity function (dexterity), and capability to carry out activities of daily living [2]. Physical performance was measured in our study with functional physical performance tests as described in the manuscript [3,4].

It is now revised to (pg 2., line 74-80): »Physical performance (regarded here as a multidimensional concept including lower extremity function (mobility), upper extremity function (dexterity), and capability to carry out activities of daily living)  was measured objectively with functional physical performance tests. We measured lower limb strength with 10 repetition sit-to-stand test [29], the aerobic capacity by the six-minute walk test [30], and hand-grip strength [31] assessed with calibrated hydraulic hand dynamometer (JAMAR 5030J1, Patterson Medical, Warrenville, Ilinois) at baseline, after eight weeks, and after 16 weeks of the intervention.«

  1. REMARK - Page 2, line 74: Please describe the explanation of "functional exercise training". Is it the same training against each subject? What is the condition of training completion?

Our response: We have added description of the explanation in the manuscript, please see pg 2., lines 85 – 99. The following text has been added:

“The functional exercise simulates activities of daily living [8] and targets the neuromuscular system to train movements that activate the nervous system and the muscle groups [9]. Functional exercise is performed as a combination of lower and upper body movements, along with various multi-joint activities [10]. The functional exercise in our study lasted for up to 30 minutes before each HD session. The number of repetitions, sets, and the load was individualized to achieve the desired intensity of the rate of preserved exertion (RPE) of 7th to 8th grade on a modified Borg scale (range 0 to 10). We started with about five different exercises with ten repetitions in two sets without the extra weight. Then, we gradually increased the number of repetitions, or we added weight. We aimed to achieve the completion of three sets of each exercise with 10-15 repetitions. The warm-up consisted of light cardiovascular exercises and coordination and balance exercises. The main part of the functional exercise consisted of varieties of squats, lunges, push-ups, pulls, pushes, and lifts tailored to each individual's abilities. The cool-down included light cardiovascular exercises alongside with stretching.The exact exercise content is additionally described in ref. 32 (Supplementary table S2).”

  1. REMARK - Page 2, line 75: Please describe the explanation of "additional intradialytic cycling exercise". Is it the same training against each subject? What is the condition of training completion?

Our response: It is now explained -please see pg. 3, line 99 – 105. The following text has been added:

“Intradialytic cycling was performed in the first two hours of HD procedure. It was supervised by the kinesiologist who aimed to continuously progress the cycling resistance or time to maintain the RPE of 4th to 5th grade on a 10-grade Borg scale. Initial intradialytic cycling duration was set to 15 minutes with a gradual increase to reach the duration of up to 60 minutes. An increase in resistance and duration was individualized according to each patient's RPE response and their motivation. This exercise prescription strategy was the same for experimental and for the control group.”

  1. REMARK - Page 2, line 75: What is the "pre-dialysis functional training"?

Our response: Predialysis functional exercise is a term for the above-mentioned functional exercise (line 85-99). We say predialysis because it was performed before the start of the dialysis procedure.
We have now explained that in the text, pg.3, line 85: »The experimental group engaged in guided functional exercise before each HD session (predialysis functional exercise) and additional intradialytic cycling exercise (during dialysis).”

  1. REMARK - Page 2, line 76, Please describe the explanation of "functional exercise". Is it the same training against each subject?

Our response: Functional exercise means the same as functional training. We now changed all expressions to functional exercise only. The explanation is described in the text, line 85-99, please see the answer to remark no. 5.

  1. REMARK - Table 2: Please describe the reason why exclusion criteria were excluded from the experiment.

Our response: For patients with the presence of these conditions (exclusion criteria), the exercise would present a medical risk for exercise related or provoked health complications or, with some of the diagnoses such as psychotic illnes or a mental disability, would present an inability of patients to properly follow exercise guidance, especially in the second phase of the study.

  1. REMARK - Page 4, line 75: Please describe the explanation of "home-based functional exercise". Is it the same training against each subject? What is the condition of training completion?

Our response: In the second study phase, they continued with functional exercise at their homes. The exercise was still individualized, and the intensity was advised to be the same as in the first phase, where exercise was guided by the kinesiologist. We explained that in pg. 3/4, lines 126 -134:

“Additionally, experimental group patients were transferred to an unsupervised, home-based exercise functional exercise in the second phase of the study. They were advised, monitored, and motivated to perform the functional exercise at home, three times a week. On dialysis days, we assessed adherence and discussed the issues of home functional exercise. We gave them a written individualized exercise program with exercise descriptions and illustrations and discussed how to perform them in their home environment. At each dialysis session of study phase 2, they reported the details of the exercise performed. The exercise was counted as completed if they maintained the same RPE level and implementation of suggested exercises as in the first guided phase of the study.”

  1. REMARK - Page 5, 2.3: Please describe all questionnaires of DB and OEE to understand the experiment.

Our response: We described the questionnaires by listing all the questionnaire items for better understanding, please see pg. 4 and 5., lines 154-166 and 176-180 for DB and OEE, respectively:

Pg.4, lines 154-166 for DB questionnaire: »The pros subscale includes the following items: “I would have more energy for my family and friends if I exercised regularly”; Regular exercise would help me relieve tension”; I would feel more confident if I exercised regularly”; I would sleep more soundly if I exercised regularly”; I would feel good about myself if I kept my commitment to exercise regularly”; I would like my body better if I exercised regularly”; It would be easier for me to perform routine physical tasks if I exercised regularly”; I would feel less stressed if I exercised regularly”; I would feel more comfortable with my body if I exercised regularly”; Regular exercise would help me have a more positive outlook on life”. The items in the cons subscale are: “I think I would be too tired to do my daily work after exercising”; “I would find it difficult to find an exercise activity that I enjoy that is not affected by bad weather”; “I feel uncomfortable when I exercise because I get out of breath and my heart beats very fast”; “Regular exercise would take too much of my time”; “I would have less time for my family and friends if I exercised regularly”; “At the end of the day, I am too exhausted to exercise”.”

Pg. 5, lines 176-180 for OEE questionnaire: »The following nine statements are included:Makes me feel better physically; Makes my mood better in general; Helps me feel less tired; Makes my muscles stronger; Is an activity I enjoy doing; Gives me a sense of personal accomplishment;Makes me more alert mentally; Improves my endurance in performing my daily activities; and Helps to strengthen my bones

  1. REMARK - Page 5, line 192: Please describe the formula of Cronbach alpha.

Our response: We decribed the formula used, please see pg.5, line 191.    (N = the number of subjects;  = average covariance between pairs;  = average variance)

  1. REMARK - Page5, line 138: The adherence was not the ratio but the total number? Formula helps readers understand.

Our response: We added the formula, please see Pg. 7, Line 187: »Patient’ adherence (%) = number of completed sessions / number of offered sessions * 100«

  1. REMARK - Page 5, line 158: Please describe the definition of "Absolute baseline scores".

Our response: To avoid the confusion of readers we have changed the term to »Baseline scores«. These are just the plain questionnnaire scores  as calculated from the questionnaires and presented in Figure 2. Previous term »absolute« was only given to stress that these numbers were not any changes (or relative) scores.

  1. REMARK - Page 6, lines 167-169: Please describe the formula of the univariate general linear model to clarify the explanatory and objective variables.- Table 3: Please explain B, 95% CI, p and Partial Eta Squared.

Our response: The following descriptions and explanations were added to legend, table 3: »Associations were tested by the univariate general linear model whereby Y(dependant variable) = B (coefficient) x X (independent variable) + U (error); B, independent variable model coefficient; 95% CI, 95% confidence interval for B; p, statistical probability; partial eta squared, measure of effect size representing the proportion of total variance of dependant variable associated with the independent variable.«

  1. REMARK - Page 8, line 199: Does "cycling sessions" mean "additional intradialytic cycling exercise" in line 75? If so, please unify it.

Our response: Yes, »cycling sessions« refer to intradialytic cycling exercise. We changed it to cycling exercise only. In the text, we have somewhere put the expression »intradialytic« before cycling exercise, so that it is clearly evident to the reader that the cycling exercise was performed during the dialysis.

  1. REMARK - Page 8, line 199: Does "in-center functional training" mean "functional exercise training" in line 75? If so, please unify it.

Our response: Yes, »in-center functional training« do refer to functional exercises. In our study, there are two types of functional exercise – one is performed in the dialysis center (in-center) before the dialysis (predialysis, first study phase) and the other one at home (second study phase). That is why we had to divide them with these expressions before the word functional exercise. We unified the word functional exercise/training to functional exercise only. However, somewhere we kept the preposition predialysis and home-based for better clarity and understanding. 

  1. REMARK - Page 8, lines 194-195: In the higher and lower group of OEE and DB, how was the composition ratios of the experimental and control group? Please describe it.

Our response: We have now added the composition ratios of the EXP and CON group. Please see page 8, lines 275 – 279: »In the OEE high group, there were nine patients from the EXP group and 11 patients from the CON group. OEE low group consisted of 11 EXP and 9 CON patients. In the DB high group, there were 12 patients from the EXP group and ten patients from the CON group. In the DB low group, there were 8 EXP patients and 10 CON patients.«

  1. REMARK - Page 10, line 232: Please describe how the authors concluded that OEE scale can be used for predicting the exercise adherence. Does predicting the exercise adherence mean up/down of exercise adherence or the total number (the definition of adherence described in Page 5) ? How was the error of prediction?

Our response: As evident from the table 5, higher baseline results (above median) on the OEE scale predicted significantly higher exercise adherence for cycling and functional exercise. In this sense, OEE results predict up/down of exercise adherence as given by the ratio (percentages) of completed exercise sessions relative to the total prescribed number of sessions. The error of prediction is now also given in this table. We changed the statistical method (from Mann-Whitney to t-test), so we can now also present the 95% confidence interval (and also because of the reason that data is normally distributed and reviewer 3 made a remark about using Mann-Whitney test).

  1. REMARK - As described in lines 252-260, the intervention might contribute to the development of a positive psychological state such as a stronger sense of mastery. The positive psychological state might be correlated to OEE/DB and adherence. But the intervention might also contribute to the improvements in physical performance. Even if there were correlations between OEE/DB, adherence and physical performance, these correlations might be caused by the intervention. It is difficult to judge OEE/DB could be used for predicting physical performance because the effect of the intervention can not be ignored. The further experiment is needed.

Our response: Please note, that we examined the association between baseline scores of OEE and DB and subsequent adherence and improvement of physical performance. We wanted specifically to see if we can divide patients using the baseline questionnaire scores prior to the implementation of exercise programs into more and less compliant groups concerning later adherence and into respondents and non-respondents to exercise based on their results from subsequent physical performance tests. Since we analyzed associations between baseline pre-intervention questionnaire scores and post-intervention adherence/performance, these associations could not be caused by the intervention (pre-intervention baseline questionnaire scores could not be influenced by later given intervention). It is also important to note that prior to the baseline questionnaire application, no exercise interventions of any sort were given to patients in or outside of dialysis centers.

Reference

  1. Huang, M.; Lv, A.; Wang, J.; Xu, N.; Ma, G.; Zhai, Z.; Zhang, B.; Gao, J.; Ni, C. Exercise Training and Outcomes in Hemodialysis Patients: Systematic Review and Meta-Analysis. Am. J. Nephrol. 2019, 50, 240–254.
  2. Rose, M.; Bjorner, J.B.; Gandek, B.; Bruce, B.; Fries, J.F.; Ware, J.E. The PROMIS Physical Function item bank was calibrated to a standardized metric and shown to improve measurement efficiency. J. Clin. Epidemiol. 2014, 67, 516–526.
  3. Van Lummel, R.C.; Ainsworth, E.; Lindemann, U.; Zijlstra, W.; Chiari, L.; Van Campen, P.; Hausdorff, J.M. Automated approach for quantifying the repeated sit-to-stand using one body fixed sensor in young and older adults. Gait Posture 2013, 38, 153–156.
  4. Norman, K.; Stobäus, N.; Gonzalez, M.C.; Schulzke, J.-D.; Pirlich, M. Hand grip strength: Outcome predictor and marker of nutritional status. Clin. Nutr. 2011, 30, 135–142.
  5. Bohannon, R.W.; Bubela, D.J.; Magasi, S.R.; Wang, Y.-C.; Gershon, R.C. Sit-to-stand test: Performance and determinants across the age-span. Isokinet. Exerc. Sci. 2010, 18, 235–240.
  6. Ross, R.M.; Murthy, J.N.; Wollak, I.D.; Jackson, A.S. The six minute walk test accurately estimates mean peak oxygen uptake. BMC Pulm. Med. 2010, 10, 31.
  7. Bellace, J. V; Healy, D.; Besser, M.P.; Byron, T.; Hohman, L. Validity of the Dexter Evaluation System’s Jamar dynamometer attachment for assessment of hand grip strength in a normal population. J. Hand Ther. 2000, 13, 46–51.
  8. Beckham, S.G.; Harper, M. Functional training. ACSMs. Health Fit. J. 2010, 14, 24–30.
  9. Boyle, M. Functional training for sports; Human Kinetics, 2004; ISBN 9780736046817.
  10. Brill, P.A. Functional fitness for older adults; Human Kinetics, 2004; ISBN 9780736046565.

Reviewer 2 Report

Outcome Expectations for Exercise and Decisional Balance questionnaires help predict adherence and efficacy of exercise programs in dialysis patients

Encouraging physical activity among CKD patients with sedentary lifestyle is an important topic, as well as understanding factors influencing physical activity decisions. The manuscript has potentially interesting results, but in its current form the methods and results sections are difficult to follow and some of the methodological decisions are hard to follow. For more detailed feedback, please see the comments below.

Introduction

Line 47: “The DB questionnaire that is comprised of 10 pro and six con items was developed by Marcus et al. [14]…” – Please consider if this would sit better in methods section. The sentence is also difficult to follow. Please consider rephrasing to improve readability.

Also consider introducing the Social-cognitive theory already in the introduction, not only in the discussion section. A number of references were introduced first time in the discussion section (such as Michie et al.), it would have benefited the reader if they had already been included in here.

Line 61: Aims are clearly presented. However, the reader is left wondering whether there are any clear hypothesis formulated? Please could you add your hypothesis - if there were some - or clearly explain that the research was explorative. Considering that the authors use the social cognitive theory in their research, the theoretical background would allow formulating hypothesis of the expected results.

Methods and materials

Line 63: This reads like continuing of the research aims from the introduction. Please consider combining in the introduction section.

Line 73: Patients' demographic and clinical characteristics are presented in Table 1. – This belongs to the results section.

While the methods paper has been published elsewhere, for the benefit of the reader, please add further details of the intervention design. At the moment it is difficult to understand how the intervention was designed and what it consisted of. For example, details of the physical activity intervention – what did it consist of? How often? How were the participants recruited? At this point, it is also not clear whether in this manuscript analyses are based on group allocation. This only becomes clearer in the statistical analysis section.

Statistical analyses

Considering the relatively low number of the participants, please add information of any missing data and how this was dealt with. Please, could you also add number of participants included in the different analyses. At the moment it is somewhat confusing, as the first part of the analyses are conducted separately within/between groups, but the modelling (GLM) appears to be done for all participants together.

Line 146: Mann-Whitney test was used to check for the differences in exercise adherence, where subjects were divided into two subgroups according to their score on the questionnaire: a low score subgroup (OEE or DB score below median) and a high score subgroup (OEE or DB score above median). – please could you add a sentence in clarifying why this method was selected. Alternatively, for example, participants could have been divided depending on the frequency of attended sessions.

The subgroup division needs to be also clarified as it is not clear for the reader whether the subgroups were divided “twice” – once for EEO and once for DB scores. This would also mean that the profile of the groups used for the subgroup analyses could be very different from each other – depending on the questionnaire. i.e. did participants have consistently scores below median for both EEO and DB?

Although authors use the same EEO and DB scores in multiple comparisons, the p-value has not been adjusted for multiple testing. Authors should consider doing this or explain why this hasn’t been done.

Results

Please, in addition to median scores, could you add mean scores for OEE and DB scales – e.g. in the figure 3 more precisely.

Line 170: Do the results mean that - Better functioning at baseline was associated with higher EEO & DB scores? Please, could you clarify in the text.

In addition, the reader was left to wonder that while socio-demographic variables were similar between the randomised groups, were the low-high median score (EEO/DB) sub-groups similar in socio-demographic characteristics. Please consider adding some information – also in the discussion section.

Discussion

As noted in previously, authors should consider introducing some of the references in the discussion already in the introduction section. Especially theoretically relevant references should be discussed earlier.

Limitations of research results should also be expanded. While authors are clear that the results apply e.g. for the lower median scores within the current sample, there may be some confusion in describing a median score of 3.15 as a low score. Self-selection bias?

Author Response

Reviewer 2

Encouraging physical activity among CKD patients with sedentary lifestyle is an important topic, as well as understanding factors influencing physical activity decisions. The manuscript has potentially interesting results, but in its current form the methods and results sections are difficult to follow and some of the methodological decisions are hard to follow. For more detailed feedback, please see the comments below.

Our response: Thank you for your detailed review of our manuscript and for providing some insightful and thought-provoking suggestions to strengthen our manuscript. We feel we have sufficient responses to each of your major concerns listed above, which are further detailed below, and hope that they alleviate the concerns you have regarding the approaches adopted in our manuscript.

Introduction

  1. REMARK - Line 47: “The DB questionnaire that is comprised of 10 pro and six con items was developed by Marcus et al. [14]…” – Please consider if this would sit better in methods section. The sentence is also difficult to follow. Please consider rephrasing to improve readability.

Our response: Thank you for your suggestion. We removed the sentence from the introduction. The reference is already mentioned in the methods section.

  1. REMARK - Also consider introducing the Social-cognitive theory already in the introduction, not only in the discussion section. A number of references were introduced first time in the discussion section (such as Michie et al.), it would have benefited the reader if they had already been included in here.

Our response: We agree with your comment. Therefore, we have added the studies about behavior changes in the introduction part. Plee see page 2, lines 52-57.

  1. REMARK - Line 61: Aims are clearly presented. However, the reader is left wondering whether there are any clear hypothesis formulated? Please could you add your hypothesis - if there were some - or clearly explain that the research was explorative. Considering that the authors use the social cognitive theory in their research, the theoretical background would allow formulating hypothesis of the expected results.

Our response: We presented this after the aim of the paper, as requested, please see pg 2., Lines 67-69: »We hypothesized that higher results on OEE and DB questionnaires associate with higher adherence with and larger improvement of physical exercise intervention.«

Methods and materials

  1. REMARK - Line 63: This reads like continuing of the research aims from the introduction. Please consider combining in the introduction section.

Our response: As suggested, we have moved some parts in the introduction and others in the experimental program, the introduction section now reads as (page 2, lines 62-69): »Therefore, we aimed to define if a baseline score on OEE and DB scale can predict the adherence and success of dialysis patients in an organized exercise program, which of the two instruments performs better, and how do they perform in monitoring the change of attitudes toward exercise. We measured the baseline and subsequent change in patients' attitudes toward exercise using OEE and DB questionnaires after eight and 16 weeks of participation in a randomized controlled trial of two dialysis-bundled exercise programs. We hypothesized that higher results on OEE and DB questionnaires associate with higher adherence with and larger improvement of physical exercise intervention.«

  1. REMARK - Line 73: Patients' demographic and clinical characteristics are presented in Table 1. – This belongs to the results section.

Our response: We have now moved it to the result section. Please see line 216.

  1. REMARK - While the methods paper has been published elsewhere, for the benefit of the reader, please add further details of the intervention design. At the moment it is difficult to understand how the intervention was designed and what it consisted of. For example, details of the physical activity intervention – what did it consist of? How often? How were the participants recruited? At this point, it is also not clear whether in this manuscript analyses are based on group allocation. This only becomes clearer in the statistical analysis section.

Our response: We added more details about the intervention in the methods section, describing the details and contents of the exercise intervention. Please see pg. 2, lines 85-105 and the answers to reviewer 1 remarks 5 and 6.

Statistical analyses

  1. REMARK - Considering the relatively low number of the participants, please add information of any missing data and how this was dealt with. Please, could you also add number of participants included in the different analyses. At the moment it is somewhat confusing, as the first part of the analyses are conducted separately within/between groups, but the modelling (GLM) appears to be done for all participants together.

Our response:

(1) Baseline scores presented in Figure 2 have been analyzed for the total sample and this is evident from the x-axis designation (»total sample«).

(2) Associations between baseline questionnaire results (dependant variable) and possible predictive parameters were done in the total sample, as stated now in the legend to the table 3: »These analyses were done in the total sample n=40.«

(3) progression of changes of attitudes to exercise as captured by questionnaires are presented in figure 3 and clearly designated as divided by experimental and control groups (black and grey bars).

(4) For adherence prediction by the questionnaires, this was done with all patients of EXP and CON group together for cycling adherence and in EXP group for functional exercise adherence only (as designated by * in the legend to the table 5: »*Calculated for experimental group only«. The exact number of patients successfully finishing the first and second study phase and included in analyses is slightly lower than 40 and 20, due to drop-outs from the study; the exact numbers are given in the first column of table 5 in the parentheses.

(5) For the association between baseline questionnaire scores and final physical improvement after 16 weeks, all patients successfully finishing both study phases were included; see legend to Table 6: »N=34 for all analyses (all patients successfully finishing both study phases included).«

  1. REMARK - Line 146: Mann-Whitney test was used to check for the differences in exercise adherence, where subjects were divided into two subgroups according to their score on the questionnaire: a low score subgroup (OEE or DB score below median) and a high score subgroup (OEE or DB score above median). – please could you add a sentence in clarifying why this method was selected. Alternatively, for example, participants could have been divided depending on the frequency of attended sessions.

Our response: Please note, that so far, there is a lack of recognized and validated thresholds of results of these two questionnaires. So any investigator-imposed division of the sample to subgroups can introduce bias in the analysis unless it is done by some methodologically unbiased division. One of the most frequently used approaches to avoid this »division bias« in the scientific literature is to divide the sample according to median (or tertiles, quartiles, quintiles, as appropriate). In our rather limited sample we have chosen to divide the patients in low and high questionnaire score groups according to the median not to loose statistical power with a more distributed division (e.g. to tertiles or quartiles).
Please note that due to the demands of reviewer one (that we fully recognize) the need to present an error for the adherence predictive estimate of questionnaires was presented. To this purpose, we have calculated 95% confidence interval for the difference in adherence between high and low questionnaire groups as an error estimate. With these statistics, it is correct to report a t-test p-value, so we replaced the Mann- Whitney test originally used here with the t-test. The assumption of normality distribution was not violated.

Please see page 5, Line 198-202: »Independent samples t-test was used to check for the differences in exercise adherence, where subjects were divided into two subgroups according to their score on the questionnaire: a low score subgroup (OEE or DB score ≤ median) and a high score subgroup (OEE or DB score > median). Division by the median was chosen to avoid any investigator imposed bias in choice of the division threshold.«

  1. REMARK- The subgroup division needs to be also clarified as it is not clear for the reader whether the subgroups were divided “twice” – once for OEE and once for DB scores. This would also mean that the profile of the groups used for the subgroup analyses could be very different from each other – depending on the questionnaire. i.e. did participants have consistently scores below median for both OEE and DB?

Our response: We divided patients twice. For the OEE analysis, we divided them according to the OEE median score and for DB analysis by the DB median score. Since we are interested in the predictive performance of a specific questionnaire, this is the most logical method.
Please see page 8, line 275: »Patients were divided twice, according to OEE, and to DB median score«

  1. REMARK - Although authors use the same OEE and DB scores in multiple comparisons, the p-value has not been adjusted for multiple testing. Authors should consider doing this or explain why this hasn’t been done.

Our response: Since this was a first exploratory analysis of OEE and DB questionnaire performance as exercise adherence predictors in a limited sample no p-value adjustments for multiple comparisons were done, this explanation was added to methods section, pg. 5, Line 209. 

Results

  1. REMARK - Please, in addition to median scores, could you add mean scores for OEE and DB scales – e.g. in the figure 3 more precisely.

Our response: We added a new table with mean scores for both questionnaires and groups. We also added effect sizes to demonstrate the magnitude of change. See Table 4.

  1. REMARK - Line 170: Do the results mean that - Better functioning at baseline was associated with higher OEE & DB scores? Please, could you clarify in the text.

Our response: Yes, it means that patients who had higher baseline OEE score were functioning better at baseline (6-minute walk test and handgrip-strength test) and had higher baseline LTI.
We clarified this in the text, page 6, line 234-237: »The model results showed that the OEE baseline score is significantly positively associated with a baseline result on a 6MWT (p = 0.001), in HG (p = 0.024) and with the baseline LTI (p = 0.033). This means that better functioning at baseline and higher LTI were associated with a higher OEE score. This means that better functioning at baseline and higher LTI were associated with a higher OEE score.«

  1. REMARK - In addition, the reader was left to wonder that while socio-demographic variables were similar between the randomised groups, were the low-high median score (EEO/DB) sub-groups similar in socio-demographic characteristics. Please consider adding some information – also in the discussion section.

Our response: We have checked and there were no significant differences found in age, sex distribution or dialysis vintage between high and low OEE and DB questionnaire groups. This was added to results section, please see page 6, line 223.

Discussion

  1. REMARK - As noted in previously, authors should consider introducing some of the references in the discussion already in the introduction section. Especially theoretically relevant references should be discussed earlier.

Our response: We have added the references dealing with social cognitive theory in the introduction part. See page 2, lines 52-57.

  1. REMARK - Limitations of research results should also be expanded. While authors are clear that the results apply e.g. for the lower median scores within the current sample, there may be some confusion in describing a median score of 3.15 as a low score. Self-selection bias?

Our response: Yes, we agree that the limit of 3.15 may not be externally generalizable to other, non-dialysis populations. Please note, that this was the first exploratory analysis in a somewhat limited sample. So the usual precautions about external generalization of these results apply. We have added this limitation to the discussion section, please see page 11., lines 369-377:

»Although the findings of the present study are encouraging, there are several limitations. One limitation which is common in psychological research is that the scores representing behavior changes were based on self-report. Moreover, the small sample size could be considered as a limitation, although we did find some clearly significant results. Specifically, the thresholds for division into high and low OEE and DB questionnaire scores may not be generalizable to other, especially non-dialysis populations. As this was a first exploratory analysis of these questionnaires for adherence prediction, usual precautions about external generalization of these results must be considered regarding not only the sample size but also regarding the nature of the sample, sex, age variability, and other sociodemographic variables that may be considered in future studies.«

Reviewer 3 Report

General comments

This is an interesting and worthwhile study on exercise adherence in people undergoing dialysis. The article is clearly written, and provides a useful addition to the literature.

Specific comments

Introduction

  • Which self-efficacy scale are you referring to in line 51? This is the only time self-efficacy is mentioned in the introduction.

Methods

  • Presumably the scores from the OEE and DB questionnaires were not normally distributed, which is why the Mann-Whitney test was performed. This needs to be stated in the methods section. It would useful to justify the choice of the median to divide participants onto low and high score groups due to the lack of thresholds for the two questionnaires.
  • Line 133 - change divide to dividing

Results

  • Line 158 This should be Figure 2 rather than Figure 1
  • Figure 2: Why are the results presented in this way? What does the x-axis signify here?
  • According to Figure 2, it looks as though 4 participants are straddling the median for the DB scores. What groups were these participants put into?
  • Line 186: What were the magnitudes of the between-group differences? Only the p-values are reported. It would be worthwhile reporting the effect sizes such as Cohen’s d for the progression of the attitudes to exercise, rather than just the differences.

Discussion

  • The adherence rates of the current study are much higher than those reported in the introduction of 30-40%. It would be worthwhile addressing potential reasons for this in the discussion.

Author Response

General comments

This is an interesting and worthwhile study on exercise adherence in people undergoing dialysis. The article is clearly written, and provides a useful addition to the literature.

Our response: Thank you for your detailed review of our manuscript and for providing some insightful and thought-provoking suggestions to strengthen our manuscript. We feel we have sufficient responses to each of your major concerns listed above, which are further detailed below, and hope that they alleviate the concerns you have regarding the approaches adopted in our manuscript.

Specific comments

Introduction

  1. REMARK - Which self-efficacy scale are you referring to in line 51? This is the only time self-efficacy is mentioned in the introduction.

Our response: We have mentioned the self-efficacy scale that was presented in the reference by Pinto et al (12). We agree that it may be confusing to readers since it is not a subject of our research. So the sentence has been changed and only findings relevant to hereby used questionnaires stated, please see pg 1, line 44: »According to Pinto et al. [12], decisional balance (DB) is a psychological construct that has received much empirical research. The DB, as defined by Plotnikoff [13], involves the perceived advantages (pros) and disadvantages (cons) of continuing a current behavior or adopting a new behavior.«

Methods

  1. REMARK - Presumably the scores from the OEE and DB questionnaires were not normally distributed, which is why the Mann-Whitney test was performed. This needs to be stated in the methods section. It would useful to justify the choice of the median to divide participants onto low and high score groups due to the lack of thresholds for the two questionnaires.

Our response: Please note that due to the demands of reviewer one (that we fully recognize) the need to present an error for the adherence predictive estimate of questionnaires was presented. To this purpose, we have calculated 95% confidence interval for the difference in adherence between high and low questionnaire groups as an error estimate. With these statistics, it is correct to report a t-test p-value, so we replaced the Mann- Whitney test originally used here with the t-test. The assumption of normality distribution was not violated.

Also, please note, as you correctly commented above, so far, there is a lack of recognized and validated thresholds of results of these two questionnaires. So any investigator-imposed division of the sample to subgroups can introduce bias in the analysis unless it is done by some methodologically unbiased division. One of the most frequently used approaches to avoid this »division bias« in the scientific literature is to divide the sample according to median (or tertiles, quartiles, quintiles, as appropriate). In our rather limited sample we have chosen to divide the patients in low and high questionnaire score groups according to the median not to loose statistical power with a more distributed division (e.g. to tertiles or quartiles). Please see page 5, Line 198-202: »Mann-Whitney test was used to check for the differences in exercise adherence, where subjects were divided into two subgroups according to their score on the questionnaire: a low score subgroup (OEE or DB score ≤ median) and a high score subgroup (OEE or DB score > median). Division by the median was chosen to avoid any investigator imposed bias in choice of the division threshold.«

  1. REMARK - Line 133 - change divide to dividing

Our response: Thank you for your comment. We have changed the word to dividing (pg.6, line 180).

Results

  1. REMARK - Line 158 This should be Figure 2 rather than Figure 1

Our response: Thank you for noticing this mistake. We have now changed it to Figure 2.

  1. REMARK - Figure 2: Why are the results presented in this way? What does the x-axis signify here?

Our response: We have used the distributed scatter plot in Figure 2 to represent to the reader in a clear and concise (space saving) manner the distribution of score values together with information about the relation of this distribution to the sample median, which was used to divide the sample for further analysis. It also gives the reader an insight into a frequency distribution. Please note that due to the demands of reviewer 2, the numerical results are now also presented in the Table 4. Distributed scatter plot is a very common way of result presentation, especially in exploratory studies of variables in unstudied populations such as ours.

  1. REMARK - according to Figure 2, it looks as though 4 participants are straddling the median for the DB scores. What groups were these participants put into?

Our response: Thank you for pointing out this question. We clarified this in the statistical analysis section (Methods). Please see page 5, line 200: »a low score subgroup (OEE or DB score ≤ median) and a high score subgroup (OEE or DB score > median).« So the cases with exact median values belong to the low score group.

  1. REMARK - Line 186: What were the magnitudes of the between-group differences? Only the p-values are reported. It would be worthwhile reporting the effect sizes such as Cohen's d for the progression of the attitudes to exercise, rather than just the differences.

Our response: Thank you for your suggestion. In Table 4 we report numerical mean ± SD values and effect sizes to better present the progression and magnitudes of the attitudes to exercise.
In statistical analysis, we explained how we calculated the effect sizes. See page 5, line 195:
»Cohen d effect sizes (ES) were calculated to determine the magnitude of the group differences in OEE and DB scores. ES was classified as: <0.2 was defined as trivial, 0.2 - 0.6 as small, 0.6 – 1.2 as moderate, 1.2 – 2.0 as large. And >2.0 as very large [33]

Discussion

  1. REMARK - The adherence rates of the current study are much higher than those reported in the introduction of 30-40%. It would be worthwhile addressing potential reasons for this in the discussion.

Our response: In the introduction, we mentioned the attrition level (withdrawal from the study) to be around 30-40%, which is only an indirect implicit measure of adherence. It may also be subjected to influences other than poor adherence, such as intercurrent illness. Nevertheless, adherence in our study was indeed high. We addressed the potential reasons for that in the discussion. Please see page 11, line 365-368: »We believe that constant personal contact of the kinesiologist and nurses combined with strong support from physicians was the key to high exercise adherence in our study. Additionally, close family members were informed about the study and welcomed to support the patient.«

Round 2

Reviewer 1 Report

All of my comments were answered.
This paper became very clear to me, and it should be accepted and published in our journal.
Thank you for your submission and correction.

Reviewer 2 Report

Thank you for the careful consideration of the comments made in the previous manuscript version. Regarding the changes made and the author Feedback, I have no further comments and am happy to recommend the manuscript for publication.